# Macroscopic Poly Schiff Base-Coated Bacteria Cellulose with High Adsorption Performance

**DOI:** 10.3390/polym12030714

**Published:** 2020-03-23

**Authors:** Lili Ren, Zhihui Yang, Lei Huang, Yingjie He, Haiying Wang, Liyuan Zhang

**Affiliations:** 1School of Metallurgy and Environment, Central South University, Changsha 410083, China; ren_lili@csu.edu.cn (L.R.); yangzh@csu.edu.cn (Z.Y.); znhuanglei@csu.edu.cn (L.H.); yingjiehe@csu.edu.cn (Y.H.); 2Chinese National Engineering Research Center for Control and Treatment of Heavy Metal Pollution, Central South University, Changsha 410083, China; 3Department of Colloid Chemistry, Max Planck Institute of Colloids and Interfaces, 14476 Potsdam-Golm, Germany; 4Environmental Engineering Research Centre, Department of Civil Engineering, The University of Hong Kong, Pokfulam, Hong Kong, China

**Keywords:** bacteria cellulose, poly Schiff base, chromium, adsorption

## Abstract

Here, a nanofiber-exfoliated bacteria cellulose aerogel with improved water affinity and high mass transfer was synthesized. Consequently, poly Schiff base can be uniformly coated within the body of bacteria cellulose aerogel without the traditional dispersion treatment. The composite aerogel has adequate mechanical and thermal stability and high mass transfer efficiency. Such an aerogel can serve as a superior adsorbent for flow through adsorption of pollution. Typically, the adsorption capacity towards Cr(VI), Cu(II), Re(VII), Conga red, and Orange G reaches as high as 321.5, 256.4, 153.8, 333.3, and 370.3 mg g^−1^, respectively. Moreover, the adsorption by this composite aerogel is very fast, such that, for example, at just 2 s, the adsorption is almost finished with Cr(VI) adsorption. Moreover, the composite aerogel exhibits a good adsorption-desorption capability. This research will hopefully shed light on the preparation of bacteria cellulose-derived macroscopic materials powerful in not only environmental areas, but also other related applications.

## 1. Introduction

Hexavalent chromium (Cr(VI)), a typical toxic contaminant, is normally discharged by industrial plants, which has strong teratogenicity and carcinogenicity [1,2,3]. Various technologies have been developed and, among them, adsorption is a popular choice for Cr(VI) owing to its feasibility of operation and cost effectiveness [4,5,6,7,8,9,10,11]. The adsorbent is very important in the adsorption technique, but the current adsorbent mainly suffers from the trade-off between mass transfer and recoverability [12,13,14,15]. Typically, high mass transfer requires a good dispersion of adsorbent, but this would decline the recovery efficiency. While high recoverability needs relatively strong noncovalent interaction between adsorbent building blocks, which in turn brings down mass transfer.

Bacteria cellulose (BC), a naturally nanofiber-arranged hydrogel, is a promising candidate adsorbent, exhibiting high mechanical strength and chemical stability [16,17,18,19,20,21,22,23]. Moreover, the highly porous network is of high advantage in boosting the mass transfer within bacteria cellulose. Most recently, a polymer with rich functional groups was adopted to modify the bacteria cellulose to improve the adsorption performance. For example, Jin, X et al. and Wang, J et al. used polyethyleneimine to graft on the bacteria cellulose skeleton for the adsorption of cations (e.g., Cu(II), Pb(II), Hg(II)) [24,25]. Jahan, K et al. and Yang, Z et al. reported the in-situ fabrication of poly(aromatic amine) on bacteria cellulose for effectively separating Cr(VI) from the aqueous solution via reduction and chelation [26,27]. Unfortunately, bacteria cellulose must be dispersed firstly to allow the ultimate interaction between cellulose and polymers [24,27]. This complicates the procedures and moreover weakens the mechanical strength of the final macroscopic product [28]. Moreover, though the batch adsorption performance of these adsorbents were proven to be fine, the dynamic adsorption using macroscopic adsorbent was satisfied to practically purify continuous-flow industrial wastewater, which was scarcely reported.

Here, we report a facile fabrication of bacteria cellulose uniformly coated with poly Schiff base. The pretreatment by sodium silicate was applied first to modify the bacteria cellulose to increase the mass transfer efficiency without breaking the macroscopic structure. As a result, the polymerization of Schiff base proceeded effectively within the body of bacteria cellulose to achieve a uniform coating of polymer. The macroscopic composite monolith exhibits attracting performance in adsorption of Cr(VI) and other typical pollutants in batch and dynamic adsorption tests.

## 2. Experimental Section

### 2.1. Materials

The bacterial strain, Gluconacetobacter hansenii (ATCC 53582), was provided by China Center of Industrial Culture Collection. The m-phenylenediamine was purchased from Aladdin Industrial Co., Ltd. (Hangzhou, China). Glutaraldehyde was obtained from Macklin Biochemical Technology Co., Ltd. (Shanghai, China) and other chemical regents were provided from Sinopharm chemical reagent (Shanghai, China).

### 2.2. Preparation of Porous Bacteria Cellulose/Poly Schiff Base

Pretreatment

The bacteria cellulose was treated with the solution of alkaline sodium silicate. Then, HCl was added and sodium silicate transformed to NaCl and SiO_2_. After the removal of SiO_2_ by NaOH, the pretreated bacteria cellulose was freeze-dried and used for polymer coating.

Polymer Coating

The HCl-protonated m-phenylenediamine was mixed with glutaraldehyde, which formed the oligomer solution due to protonation. The pretreated bacteria cellulose (pBC) was immersed in precursor mixture to absorb the oligomers into matrix of pBC. The poly Schiff base was deposited on cellulose nanofibers by tuning the pH to base. The detail of procedure was given in Appendix A.

The relative content of poly Schiff base in pBC was estimated according to the following equation:(1)$$D\%=\frac{W_{product}-W_{BC}}{W_{product}}\times 100\%$$
where the D% is the content percentage of poly Schiff base in the composites, *W*_product_ is the total mass of composites, W_BC_ is the mass of bacteria cellulose or porous bacteria cellulose.

### 2.3. Characterization

The attenuated total reflection infrared (ATR) spectrum was recorded on a Nicolet IS50 Fourier transform infrared (FTIR) spectrometer (Thermo Fisher scientific instruments, Waltham, MA, USA) at resolution of 4 cm^−1^ in the range of 400~4000 cm^−1^. A field emission scanning electron microscopy (SEM, JSM-6360LV, JEOL, Tokyo, Japan) and transmission electron microscopy (TEM, JEOL-2011, Tokyo, Japan) was employed to inspect the morphology. The surface area and porosity of samples were calculated by Brunauer-Emmett-Teller (BET) and Barrett-Joyner-Halenda (BJH) method, respectively (analysis gas: nitrogen; analysis time: 670 min; thermal delay: 30 s; outgas time: 12 h; bath temperature: 50 °C and outgas temperature: 150 °C) (Autosorb iQ, Quantachrome, Ashland, Virginia, USA) [29]. The crystallographic analysis of aerogels was performed on X-ray diffractometer (XRD) with Cu-Ka radiation (D/Max-RB diffractometer, Rigaku, Tokyo, Japan) from 5~50° by using the dried film of samples. The chemical composition was characterized on a K-Alpha 1063 X-ray photoelectron spectroscope (XPS) with Al Kα X-ray as the excitation source (Thermo Fisher scientific instruments, Waltham, MA, USA).

### 2.4. Batch Adsorption

A series of adsorption experiments were conducted in the polyethylene vials at 30 °C under 150 rpm rotary shaking. 15 mg adsorbent was accurately weighed and added into the 50 mL Cr(VI) containing solution. The blank assay was carried out under the same condition without any adsorbent simultaneously. The adsorbents were filtrated from the solution by filter paper (pore size ~0.45 μm) after completion of adsorption. The analysis method of Cr(VI) concentration (and other pollutants) was given in Appendix A. Considering the chemical state of Cr, the concentration of Cr(III) can be calculated as Cr(III) = Cr_total_ − Cr(VI). To survey the effect of pH on adsorption capacity, the initial pH of solution was adjusted from 0~6 by 2 M HCl or 2 M NaOH. The adsorption isotherm experiment was implemented at pH 2 for 6 h with the initial Cr(VI) concentration varied from 100~500 mg L^−1^. For kinetics experiment, Cr(VI) solution with concentration of 400 mg L^−1^ was experienced 1~240 min adsorption at pH 2. The concentration of Cu(II), Re(VII), Conga red, and Orange G solution varied from 50~500 mg L^−1^ for 6 h adsorption to achieve their adsorption isotherm. The Cu(II), Re(VII) and Orange G was adjusted to pH 2 while the Conga red was adjusted to pH 5.5. The adsorption capacity for batch adsorption experiment was calculated as follows:(2)$$q_{t}=\frac{C_{0}-C_{t}}{m}V$$
(3)$$q_{e}=\frac{C_{0}-C_{e}}{m}V$$
where *q_t_* is the adsorption capacity at time t (mg g^−1^), *C_0_* is the initial Cr(VI) concentration in solution (mg L^−1^), *C_t_* is the Cr(VI) concentration in solution at time t (mg L^−1^), *q_e_* is the equilibrium adsorption capacity (mg g^−1^), *C_e_* is the equilibrium concentration (mg L^−1^), m is the weight of adsorbent (g), and *V* is the volume of solution (L).

### 2.5. Dynamic Adsorption

For the dynamic adsorption, 0.3 g adsorbent was compressed and squeezed in a glass column with length of 500 mm and ID of 8 mm. 25 mL Cr(VI) solution with concentration of 5~100 mg L^−1^ was adjusted to pH 2 and fed into the top of column using the peristaltic pump with controlling the flow velocity from 28.42~85.26 μL min^−1^ at ambient temperature. The concentration of Cr(VI) (and other pollutant) in effluent was measured by the same method as batch adsorption experiment. For other pollutant, the influent concentration of Cu(II), Re(VII), Conga red and Orange G were 50, 20, 20, and 50 mg L^−1^, respectively. The Cu(II), Re(VII), and Orange G was adjusted to pH 2 while the Conga red was adjusted to pH 5.5. The removal rate was calculated as follows:(4)$$R\%=\frac{C_{0}-C_{a}}{C_{0}}\times 100\%$$
where *R%* is the removal rate of effluent, *C_0_* is the influent concentration of pollutant (mg L^−1^), *C_a_* is the concentration of pollutant in effluent (mg L^−1^).

### 2.6. Desorption and Reusability

The recycle adsorption experiment was tested with 7 successive cycles of dynamic adsorption of Cr(VI). After each cycle, 100 mL 1 M NaOH served as the desorption solution and flowed through the column with 0.3 g adsorbent at flow velocity of 56.84 μL min^−1^. After being washed by deionized water to neutrality, the adsorbent was freeze-dried for the next cycle.

## 3. Result and Discussion

### 3.1. Pretreatment of Bacteria Cellulose

The overview of pathway to fabricate the porous bacteria cellulose/poly Schiff base was described in Scheme 1. Bacteria cellulose was saturated with the sodium silicate solution (alkaline). Then, HCl acid reacted with sodium silicate to produce SiO_2_ aggregation. The formation of SiO_2_ could enlarge the inner space of bacteria cellulose and disassemble the large bacteria cellulose fiber bundles into much thinner fibers. The SiO_2_ was removed by NaOH solution and the pure modified bacteria cellulose was obtained by water rinse and subsequent freeze-drying. After the pretreatment, the bacteria cellulose was still macro-scale monolith. As shown in Figure 1a,b, the smooth surface of pristine bacteria cellulose consisted of compact fibers. In contrast, the pretreated bacteria cellulose had many macroscopic holes (Figure 1e), which should be generated by the SiO_2_ formation. The photograph and high-magnification images of bacteria cellulose/SiO_2_ composites can verify this (Figure 1c,d). More interestingly, the original bacteria cellulose fibers with diameter 400 nm was decreased to 30 nm (Figure 1f). The small nanofibers were most possibly free from the aggregated cellulose fibers by alkaline treatment. Moreover, the SiO_2_ formation within the gap or pores in the aggregated fiber bundles would also produce small nanofibers.

XRD was used to validate the variation of bacteria cellulose before and after pretreatment. As seen in Figure 2a, the raw bacteria cellulose exhibited characteristic of typical cellulose I by diffraction peaks at 14.3° and 22.6° [30,31,32]. Noticeably, after the pretreatment, the peak intensity at 14.3° and 22.6° attenuated obviously, which indicated the decrease of crystallinity. On the other hand, the relative intensity of 16.4° increases. Moreover, a small peak located at 20.0° appeared, indexed as the (110) of cellulose II. It inferred that the structure of semi-crystalline region and amorphous region was altered via pretreatment. This can be explained by the separation of small cellulose nanofibers from original fiber bundles, which inevitably exposed new facet. It must be mentioned that in the raw cellulose the H bonding (by –OH) pushed the formation of cellulose crystal. The decrease of crystallinity of bacteria cellulose demonstrated the breaking of H bonding, which signified that the –OH is free from the crystal. This is conducive to increasing its affinity toward aqueous solution, which can promote the mass transfer within the body of bacteria cellulose.

### 3.2. Schiff Base Loading in Bacteria Cellulose

To allow the precursor coat onto the nanofibers, the bacteria cellulose firstly interacted with acidic aqueous mixture of *m*-phenylenediamine and glutaraldehyde. The pretreated bacteria cellulose immersed into precursor solution as soon as their contact. In contrast, the raw bacteria cellulose took ~30 min to be absolutely wetted by the solution. That should be caused by the pretreatment of bacteria cellulose, which alters its surface properties. After the saturation of precursor, the cellulose was put into the alkaline solution to speed up the polymerization process. Due to the strong interaction between –NH_2_ and –OH, the poly Schiff base would stably coat onto the nanofibers of bacteria cellulose. The white cellulose became brown after the polymer coating (Figure 1g), which had ultra-low density. The final product was named as pBC-Polym-*x*, where pBC is the pretreated bacteria cellulose and *x* is the concentration of m-phenylenediamine in precursor solution. For comparison, the product without pretreatment was denoted as BC-Polym-*x*.

More importantly, the difference on the wettability of raw and pretreated bacteria cellulose changed the loading mass of poly Schiff base. As shown in Figure 1h, the polymer loading on pretreated bacteria cellulose increased very fast and it took 30 min for saturation. However, for the raw bacteria cellulose, it spent ~120 min for the complete loading. Typically, the content of poly Schiff base on pBC was at least 20% higher than that on raw materials, which can be vividly distinguished by the color of the product (Appendix A). This should be related to the improvement of the wettability of bacteria cellulose by the pretreatment.

### 3.3. Characterization

The ATR-FTIR technique was applied to study the structural variation of pBC before and after polymer coating. As seen in Figure 2b and c, functional groups including –OH (3343 cm^−1^), C–H (2902 cm^−1^), and –C–O–C– (1109 and 1164 cm^−1^) can be found in the bare pBC [33,34,35]. After the polymer coating, two absorption bands at 1609 and 1493 cm^−1^ emerged. These corresponded to the stretching vibration of C=N and N–H shearing vibration in benzenoid amine structure [36,37]. Moreover, these two signals became obvious with the increase of polymer amount coating on the nanofibers.

The morphology of the final product was examined by the SEM technique, taking pBC-Polym-0.04 as an example. The morphology of the surface section was shown in Figure 3a that the gap between the nanofibers was filled with polymer. The result was identical for the cross-section of the product (Figure 3b). That is to say, the poly Schiff base uniformly coated on the nanofibers within the body of bacteria cellulose. The TEM images are given in Figure 3c (agglomeration) and 3d (single fiber). As shown, the incorporated thick polymer evolved to sheath-like structure to encapsulate the cellulose nanofibers, which was consistent with the SEM results. For pBC-Polym-0.02, only nanoparticles can be found on the nanofibers (Appendix A). On the other hand, pBC-Polym-0.08 showed a similar morphology but a higher density of coating (Appendix A). The N_2_ adsorption-desorption isotherm and porosity are shown in Appendix A, which indicated that the products were mainly composed of mesopores and macropores. In addition, the product exhibited good thermal stability (Appendix A), and this was beneficial for its practical applications.

### 3.4. Adsorption Performance

The effect of pH on the adsorption was investigated and the results were given in Appendix A. As shown, the optimum pH for adsorption was ~2, which was related to the balance between protonation of the polymer and speciation of Cr ions at different pH. In the following experiments, pH will be used without special caution.

#### 3.4.1. Adsorption Isotherm

Effect of initial concentration of Cr(VI) (100~500 mg L^−1^) on the adsorption was investigated. As shown in Figure 4a, the adsorption capacity of the series of pBC-Polym-*x* rapidly increased with the rise of Cr(VI) concentration. Then, the adsorption process tended to be saturated. Similar trend was found for the samples without pretreatment (Figure 4b). However, the adsorption performance is much better for pBC-Polym-*x*. This can be ascribed to the high polymer loading and the high affinity to aqueous solution of pBC. On the other hand, the saturated capacity of pBC-Polym-0.04 was very close to that of pBC-Polym-0.08. The possible reason was that the thicker polymer coating on the nanofibers (pBC-Polym-0.08) decreased the utilization efficiency of the active materials.

Furthermore, the experimental data (Figure 4c~f) were simulated by Langmuir and Freundlich models. The Langmuir and Freundlich equations are shown below:(5)$$\frac{C_{e}}{q_{e}}=\frac{C_{e}}{q_{m}}+\frac{1}{bq_{m}}$$
(6)$$\log q_{e}=\log K_{f}+\frac{1}{n}\log C_{e}$$
where *C_e_* is the Cr(VI) concentration at equilibrium, *q_e_* is the adsorption capacity at equilibrium, *q_m_* is the maximum adsorption capacity at saturation, *b* is the isotherm constant for Langmuir model, *K_f_* and *n* are the constants of isotherm equation. The parameters were listed in Table 1.

Based on the calculation, adsorption behavior can be better described by the Langmuir model. This demonstrated a homogenous monolayer adsorption on poly Schiff base-coated cellulose nanofibers. The maximum adsorption capacity of pBC-Polym-0.04 was 321.5 mg g^−1^. Remarkably, the adsorption capacity in this research was higher than most of the macroscopic adsorbents in previous studies (Table 2).

#### 3.4.2. Adsorption Kinetics

As shown in Figure 4g~i, the adsorption process of pBC-Polym-0.04 was very fast in the initial stage (≤30 min), involving 76% of the total capacity. This was presumably due to the physical adsorption. In this step, the negative charged Cr(VI) was adhered to positive charged adsorbent by electrostatic attraction. Additionally, the fast adsorption process was related to the high affinity to aqueous solution of the pBC-Polym-*x*, which promised effective mass transfer. The subsequently accumulative adsorption of Cr(VI) ultimately reached the equilibrium within 3 h. The pseudo-first and pseudo-second order models were adopted to analyze the adsorption process. As seen in Table 3, the pseudo-second-order model with the correlation coefficient (*R*^2^) > 0.99 indicated that the Cr(VI) adsorption was mainly controlled by chemical sorption. The adsorption process was deduced as the electrostatic attraction and redox-chelation. The Cr(VI) adsorption was initiated by electrostatic interaction. The Cr(VI) binding on surface of adsorbent was reduced to Cr(III) and chelated by quinoid imine group - the oxidation product of benzenoid amine structure [37,46]. The elaborate verification of adsorption mechanism was stated in Appendix A.

Furthermore, the Weber–Morris intraparticle diffusion model was used to describe the adsorption process (Figure 4j). The whole adsorption can be divided into three regions: (1) Cr(VI) diffusion in aqueous solution onto the exterior surface of adsorbent; (2) Cr(VI) permeation into the inner section of adsorbent and Cr(VI) diffusion into the internal polymers; (3) chemical sorption and reaching equilibrium. Clearly, the surface diffusion of pBC-Polym-0.04 completed within 1 minute. From 0~1 min, Cr(VI) was instantly captured by the poly Schiff base deposited on the surface of aerogel. In the second procedure, faster permeation and mass transfer of pBC-Polym-0.04 was observed. For BC-Polym-0.04, ascribed to the weaker affinity toward solution, the permeation and adsorption of inner polymers lasted from 5~120 min. The k_id_ listed in Table 3 manifested the high adsorption rate of pBC-Polym-0.04.

#### 3.4.3. Dynamic Adsorption

The fast adsorption process is vividly illustrated in Figure 5a. A piece of pBC-Polym-0.04 was directly immersed into the Cr(VI) solution and it was taken out after 2 s. The liquid was squeezed out that the clean water was obtained. Considering this superior property, the pBC-Polym was filled in a column to achieve flow-through adsorption. The practical image is shown in Figure 5b. Here, 25 mL of Cr(VI) solution was pumped to the column by peristaltic pump to flow through the macroscopic adsorbent with a constant velocity.

In the flow-through adsorption, effect of concentration was examined firstly. The flow velocity was set as 28.4 μL min^−1^. As shown in Figure 6a, the removal rate was persistently remained at above 96% when Cr(VI) concentration < 100 mg L^−1^. This indicated the excellent performance of pBC-Polym in dynamic process. Figure 6b revealed the relationship between influent velocity and removal efficiency in dynamic adsorption (*C*_0_ = 50 mg L^−1^). The removal rate hardly decreases with the increased flow rate. This sufficiently demonstrated the high mass transfer efficiency of the ions within the body of adsorbent and strongly verified the high prospect of pBC-Polym in wastewater purification. Moreover, the adsorbent exhibited excellent ability in the treatment of diverse metal ions and organic dyes (Appendix A). 

#### 3.4.4. Recyclability

The repeated adsorption was tested by dynamic adsorption using 25 mL of 20 mg L^−1^ Cr(VI) solution as simulated effluent. 100 mL of 2M NaOH was applied as desorption solution and flow through the adsorbent in the column. The adsorbent was rinsed and freeze-dried after desorption for next cycle. The Cr(VI) removal rate of each cycle was shown in Figure 6c. For seven cycles, no obvious performance attenuation was observed. Typically, the Cr(VI) removal rate of adsorbent remained 96.5%, implying its distinguished reusability.

## 4. Conclusion

We have successfully fabricated the poly Schiff base-coated bacteria cellulose without the conventional dispersion treatment of the cellulose aerogel. The key is to pretreat the bacteria cellulose with sodium silicate to exfoliate the nanofibers from the big bundles of raw cellulose fibers. Due to the intact texture of the pretreated bacteria cellulose, the composite aerogel is of good properties (e.g., mechanical and thermal) and high mass transfer efficiency. The adsorption capacity of the aerogel is 321.5 (Cr(VI)), 256.4 (Cu(II)), 153.8 (Re(VII)), 333.3 (Conga red), and 370.3 (Orange G) mg g^−1^. The adsorption process obeys the pseudo-second order kinetic. The adsorbent can be regenerated easily and used to remove the pollutants without obvious performance attenuation.

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
