# Peer review of "Macroscopic Poly Schiff Base-Coated Bacteria Cellulose with High Adsorption Performance"

_polymers, 2020, doi:10.3390/polym12030714_

Round 1
Reviewer 1 Report
The research work done by Haiying Wang and Liyuan Zhang co-workers “Macroscopic poly Schiff base-coated bacteria cellulose with high adsorption performance” is a good research work to study the adsorption performance . Authors documented a new process for nanofiber-exfoliated bacteria cellulose aerogel with improved water affinity. The major advantage of the present documented report is preparation of bacteria cellulose-derived macroscopic materials turns very powerful and also to other related applications. There are not many literature procedures, similar to present documented report in the literature for new developments for macroscopic poly Schiff base. Given the importance of practicality for this work, I recommend the publication of this manuscript in the Polymers.
Author Response
We appreciate the reviewer’s recommend and constructive comment. The necessary revision was made in our resubmitted manuscript.

Reviewer 2 Report
see my comments in the attached file

Author Response
We would like to sincerely give our appreciations for your valuable comments concerning our manuscript entitled “Macroscopic poly Schiff base-coated bacteria cellulose with high adsorption performance” (ID: polymers-750601). According to Editor and Reviewers′ comments, we have conscientiously made a extensive revision to our manuscript and supplemented extra data to . The comments of two nice reviewers are laid out below in italicized and specific concerns have been numbered. Revised portions are marked in red in the revised manuscript. The corrections in the paper and the responses to the comments are as follows:
Comments and Suggestions for Authors
see my comments in the attached file.
Point 1. Line 17: Perhaps the term adequate is better than the good.
Response:
We sincerely thank the reviewer for careful reading. We have replaced the “good” with “adequate” to express more precisely.
Point 2. Line 30: Full stop after references. Here and everywhere in the manuscript.
Response:
We feel sorry for our carelessness. We have thoroughly corrected this mistake both in our resubmitted manuscript and supplementary material.
Point 3. Line 84: Please give few details about BET and BJH and the relevant references.
Response:
We sincerely thank the reviewer for careful reading. The details about BET and BJH, such as analysis gas (nitrogen), analysis time (670 min), thermal delay (30 s), outgas time (12 h), bath temperature (50 ºC) and outgas temperature (150 ºC), and relevant reference was given in our resubmitted manuscript.
Point 4. Line 154: what about the semi-crystalline region of cellulose?
Response:
Thanks for the reviewer’s comment. The bacteria cellulose is a highly purified polymer with a degree of crystallinity up to 90% composed of cellulose I. The semi-crystalline region and amorphous region was unconspicuous compared with crystalline region. After pretreatment, a small peak located at 20.0 º appeared, indexed as the (110) of cellulose II. It inferred that the structure of semi-crystalline region and amorphous region was altered.
We tried our best to improve the manuscript and made some revisions. We appreciate for Editor and Reviewers' warm work earnestly. I hope the manuscript has been improved satisfactorily and that it will be accepted for publication in your journal.
Best regards.
Yours sincerely
Liyuan Zhang

Reviewer 3 Report
I’ve just finished a review of manuscript polymers-750601 titled “Macroscopic poly Schiff base-coated bacteria cellulose with high adsorption performance” by authors Lili Ren, Zhihui Yang, Lei Huang, Yingjie He, Haiying Wang and Liyuan Zhang. The proposed paper investigates adsorption, i.e. removal of Cr(VI) ions from contaminated aqueous solutions using poly Schiff base-coated bacteria cellulose.
In the paper, poly Schiff base-coated bacteria cellulose is characterized in detail, all investigations are carefully and nicely explained and each explanation is supported by appropriate reference(s).
In addition, the influence of various factors on the removal efficiency of Cr(VI) ions from aqueous solutions under static and dynamic conditions was investigated. The obtained results are also very well explained.
The proposed topic of paper is very actual in the world of science for many years. It is also fully in line with the topics of journal Polymers.
I think that the paper should be accepted for publication in the present form and with very small additions:
- English should be improved.
- Lines 243-245: It is difficult to argue that chemisorption is responsible for the rapid kinetics of pollutant removal since in this process chemical bonds are formed. Therefore, both the paper and the supplement material have provided quite sufficient explanations that can indicate and confirm that chemisorption has occurred, however, the explanation for the rapid removal of Cr(VI) ions should be reformulated and authors should emphasize physisorption.
- When explaining the Weber-Morris curve, the explanation what does it mean when the curve goes through zero is missing.
- In addition to the results presented in the paper and the exceptionally well-explanations, section 3.4.5. is superfluous and I suggest the authors to remove it from the manuscript.
Thanks to the authors for such a well-written paper and for giving me the opportunity to review it.
Best regards
Author Response
We would like to sincerely give our appreciations for your valuable comments concerning our manuscript entitled “Macroscopic poly Schiff base-coated bacteria cellulose with high adsorption performance” (ID: polymers-750601). According to Editor and Reviewers′ comments, we have conscientiously made a extensive revision to our manuscript and supplemented extra data to . The comments of two nice reviewers are laid out below in italicized and specific concerns have been numbered. Revised portions are marked in red in the revised manuscript. The corrections in the paper and the responses to the comments are as follows:
Point 1. English should be improved.
Response:
We feel sorry for our carelessness. In our resubmitted manuscript, we have improved our English writing.
Point 2. Lines 243-245: It is difficult to argue that chemisorption is responsible for the rapid kinetics of pollutant removal since in this process chemical bonds are formed. Therefore, both the paper and the supplement material have provided quite sufficient explanations that can indicate and confirm that chemisorption has occurred, however, the explanation for the rapid removal of Cr(VI) ions should be reformulated and authors should emphasize physisorption.
Response:
Thanks for the reviewer’s constructive comment. We agree that rapid removal of Cr(VI) ions should be reformulated and physisorption should be emphasized. Hence, we have corrected “This should be due to the physical and surface chemical adsorption” into “This was presumably due to the physical adsorption”. Moreover, the elaboration was made to further explicate the physisorption induced by electrostatic attraction.
Point 3. When explaining the Weber-Morris curve, the explanation what does it mean when the curve goes through zero is missing.
Response:
We sincerely thanks for your careful reading. As your suggestion, we have revised the Figure 4 and explained the adsorption process from 0~1 min: Cr(VI) was instantly captured by the poly Schiff base deposited on the surface of aerogel. In the second procedure, faster permeation and mass transfer of pBC-Polym-0.04 was observed. For BC-Polym-0.04, ascribed to the weaker affinity toward solution, the permeation and adsorption of inner polymers lasted from 5~120 minute.
Figure 4. (a) Cr(VI) adsorption isotherms of pBC-Polym-0.02, pBC-Polym-0.04 and pBC-Polym-0.08, and (b) BC-Polym-0.02, BC-Polym-0.04 and BC-Polym-0.08. Their corresponding fitted Langmuir (c) and (d) and Freundlich isotherms (e) and (f). Effect of contact time on Cr(VI) adsorption (g) and kinetics modeling: (h) pseudo-first-order kinetic plots, (i) pseudo-second-order kinetic plots and (j) plots of intra particle model.
Point 4. In addition to the results presented in the paper and the exceptionally well-explanations, section 3.4.5. is superfluous and I suggest the authors to remove it from the manuscript.
Response:
We sincerely thanks for your careful reading. As your suggestion, the section 3.4.5 was removed and accordingly, the Figure 6 was revised.
Figure 6. Effect of initial Cr(VI) concentration (a) and effect of velocity (b) on dynamic adsorption by pBC-Polym-0.02, pBC-Polym-0.04 and pBC-Polym-0.08; recycling behavior of the pBC-Polym-0.04 in Cr(VI) adsorption (c).
We tried our best to improve the manuscript and made some revisions. We appreciate for Editor and Reviewers' warm work earnestly. I hope the manuscript has been improved satisfactorily and that it will be accepted for publication in your journal.
Best regards.
Yours sincerely
Liyuan Zhang
